# Out-of-distribution Tests Reveal Compositionality in Chess Transformers

## Abstract

Chess is a canonical example of a task that requires rigorous reasoning and long-term planning. Modern decision Transformers - trained similarly to LLMs - are able to learn competent gameplay, but it is unclear to what extent they truly capture the rules of chess. To investigate this, we train a 270M parameter chess Transformer and test it on out-of-distribution scenarios, designed to reveal failures of systematic generalization. Our analysis shows that Transformers exhibit compositional generalization, as evidenced by strong *rule extrapolation*: they adhere to fundamental 'syntactic' rules of the game by consistently choosing valid moves even in situations very different from the training data. Moreover, they also generate high-quality moves for OOD puzzles. In a more challenging test, we evaluate the models on variants including Chess960 (Fischer Random Chess) - a variant of chess where starting positions of pieces are randomized. We found that while the model exhibits basic strategy adaptation, they are inferior to symbolic AI algorithms that perform explicit search, but gap is smaller when playing against users on Lichess. Moreover, the training dynamics revealed that the model initially learns to move only its own pieces, suggesting an emergent compositional understanding of the game.

## 1 Introduction

Chess has long been regarded as a symbol of human intellectual endeavor. While the most competent machine chess engines leverage highly interpretable traditional search-based algorithms Stockfish, it is also possible to learn capable chess policies directly through reinforcement learning or behavior cloning using Transformers Ruoss et al. (2024); Noever et al. (2020) and other neural networks. This raises an open question: to what extent can these black-box, model-free chess Transformers be said to truly "understand" the game?

Studying chess holds significant relevance for broader reasoning tasks, since reasoning, too, involves chaining together a sequence of logically valid inferences and requires long-term planning and strategy. This connects to the pivotal question of whether LLMs and reasoning models can develop a genuine internal model of the process of reasoning or whether they simply reproduce fragments of strategy gleaned from statistical regularities in training data Shojaee et al. (2025); Zhou et al. (2023). To empirically test such inherent understanding, we propose evaluating a model's behavior in out-of-distribution (OOD) situations. We are particularly interested in two key aspects of this evaluation:

**Rule Extrapolation:** Can the model adhere to the fundamental rules of the task by consistently producing valid moves, even in unfamiliar, out-of-distribution settings?

**Strategy Adaptation:** Can the model productively adapt its approach to reach a desirable goal state when the game's basic rules are unchanged, but altered initial conditions render its learned strategies suboptimal?

Chess is a rich and useful context to test these phenomena in: thanks to its rigid rules, games can be described in a formal language, and the validity of moves can be easily checked using readily available software . On the other hand, several game variants and puzzle types exist, providing interesting and human-interpretable out-of-distribution test environments. Our **contributions** are:

- we created a battery of chess puzzles that present out-of-distribution situations
- we trained a model-free Transformer-based chess policy using behaviour cloning reproducing the methodology of Ruoss et al. (2024), but using a training dataset more suitable for our study;

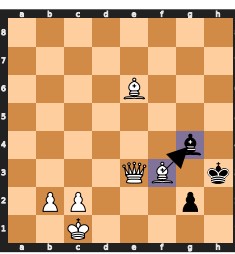 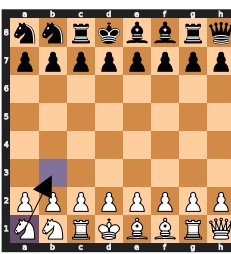 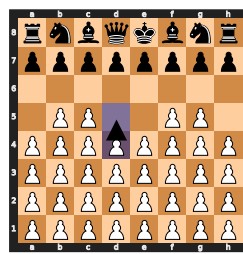

Figure 1: **OOD board types:** We consider four out-of-distribution chess scenarios. The **first** board shows a position with more pieces of a certain type than allowed (the exception is via pawn promotion), 3 white queens in this case; while the **second** one illustrates 2 white bishops on squares with the same color. We also study two chess variants: Chess960, where in the starting position pieces on the first rank are randomly reordered and this is mirrored in the 8th rank (**third** board); and Horde, where White only has pawns and Black has a different objective (**fourth** board). The next moves of our model are highlighted with purple.

- we evaluated the models on rule extrapolation and strategy adaptation;
- we evaluated the policy's ability to play full games in Chess960[1] (Fisher Random Chess), in which the starting positions of pieces are randomized, and Horde, where the Black's objective is altered;
- we analyzed the dynamics of rule learning.

## 2 BACKGROUND AND RELATED WORK

Reasoning is the process of synthesizing factual knowledge (what is known) with procedural knowledge (how to derive new information) to solve problems that are intractable with the initial facts alone. In the case of chess, using basic rules of the game together with a procedure like minimax search can yield high-quality moves in a variety of situations Stockfish. When implemented explicitly, reasoning algorithms like minimax search display a form of compositional generalization, guaranteeing high-quality solutions to a potentially infinite range of problems that adhere to the same compositional structure.

When we train language models on internet data or decision-Transformers via behaviour cloning, such reasoning-derived decisions are distilled into autoregressive neural network models. It is an open question to what degree the resulting models are able to display similar levels of compositionality and strong generalization. Empirical evidence on this question is non-conclusive (Reizinger et al. (2024)).

On the one hand, studies show that Transformer-based models display more-than-expected degrees of compositional generalization, and are able to transcend the limitations of their finite training data (Ahuja & Mansouri (2025); Han & Padó (2024); Ramesh et al. (2024); Lake & Baroni (2023)). For example, research into how language models handle formal languages—which are defined by a clear set of compositional rules—provides a controlled environment to test these abilities (Delétang et al. (2023); Mészáros et al. (2024)). In this setting, models have been shown to succeed at rule extrapolation, a challenging form of out-of-distribution generalization where they must complete prompts that violate one or more of the rules seen during training (Mészáros et al., 2024). The ability to correctly apply a subset of known rules to these novel, rule-breaking scenarios suggests that the models are not merely interpolating from training data but are learning a more abstract and flexible representation of the underlying rule system. This observation is so striking that it has led some to argue that our current understanding of statistical generalization is insufficient to explain these emergent capabilities (Reizinger et al., 2024).

This capacity for generalization is powerfully illustrated within the domain of chess—a core focus of our own work. Recent research highlights the phenomenon of transcendence, where a generative model can outperform the very experts who created its training data (Zhang et al., 2024). Specifically, a Transformer trained simply to predict moves in a large corpus of chess games was shown to achieve a higher level of play than any individual player represented in its dataset. This emergent ability arises because the model synthesizes a more robust and general strategy from diverse data sources. It effectively performs skill denoising by averaging out the individual errors of many play-

---

[1]https://en.wikipedia.org/wiki/Chess960

ers, while simultaneously achieving skill generalization by combining the specialized strengths of different experts into novel, superior strategies (Abreu et al., 2025).

Providing a more nuanced look at the Transformer architecture, Weiss et al. (2021); Zhou et al. (2023) found that a model's ability to generalize compositionally depends heavily on the underlying procedure it must learn. They investigate why Transformers succeed at some algorithmic tasks but fail at others by focusing on length generalization—the capacity to handle inputs longer than those seen during training. The authors propose the RASP-Generalization Conjecture, which posits that Transformers will successfully learn and generalize an algorithm if its solution can be expressed as a short program in RASP-L, a language designed to mirror the Transformer's native computational primitives. This framework suggests that strong generalization occurs when a task's inherent algorithm aligns well with the Transformer's architectural biases.

Other published works detailed failure cases where state-of-the-art reasoning language models fail to generalize successfully Shojaee et al. (2025); Malek et al. (2025). Closest to our work is the recent investigation by Malek et al. (2025), who find that even state-of-the-art models often fail on easier or simplified versions of tasks they otherwise excel at. This indicates that they likely memorize strategies from their training data which they aren't able to adapt to new situations, even if the new puzzle is in some sense easier to solve. This suggests that the models rely on statistical shortcuts rather than robust reasoning.

Our work aims to contribute to this body of evidence by decomposing two aspects of compositional generalization to (1) rule extrapolation - as studied by Mészáros et al. (2024); Reizinger et al. (2024), and (2) strategy adaptation: dealing with situations when some strategies memorised during training may not transfer. In this context, rule extrapolation measures whether the chess model continues to respect the rules of the game and choose valid moves even in situations it has never encountered during training, while strategy adaptation measures how competently it is at choosing a high-quality move in a game against an opponent or a puzzle. In particular, we study a chess model's ability to generate next moves for boards that are qualitatively different from the ones seen during training (see Fig. 1), and to play variants of chess including Chess960 and Horde.

## 2.1 CHESS ENGINES

Stockfish 17, currently the strongest chess engine, relies on classical search techniques like alpha-beta pruning and handcrafted evaluation functions, recently enhanced with neural network support (NNUE) for better positional insight (Nasu, 2018). In contrast, AlphaZero (Silver et al., 2017) and (LCZero, 2018) are neural network-based engines that use deep learning and reinforcement learning. AlphaZero was trained by playing millions of games against itself, using Monte Carlo Tree Search (MCTS) for decision-making. LCZero follows AlphaZero's principles but is open-source and continuously trained by a distributed network of contributors. Recently, searchless methods have emerged (Ruoss et al., 2024; Monroe & Chalmers, 2024): Transformer architectures were trained on large datasets and annotated by Stockfish, and they achieved grandmaster level. Other studies (Zhang et al., 2024; Noever et al., 2020; Toshniwal et al., 2022), used the Transformer architecture in a self-supervised way: the training set consisted only of games given by the move history. In this paper, the use Stockfish 17 (the version of May 2025) with depth 20 as an oracle, and train a searchless Transformer-based model which is capable to produce next moves for *any* board states.

## 3 EXPERIMENTAL SETUP

### 3.1 TRAINING

To study the rule extrapolation and strategy adaptation of Transformer-based chess models, we train a Transformer with $\approx 270$ million parameters using supervised learning, similarly to Ruoss et al. (2024). We train on their ChessBench dataset created for behaviour cloning. It consists of board positions with the labels as the best next move generated by Stockfish 16, in a way that each legal move from the board state was assigned a score by Stockfish (with a time limit of $50ms$), and the move with the highest score was selected as the best. Generally, there are 30 legal moves from a board state, thus Stockfish spends approximately $1.5s$ per board. We treat states reachable by pawn promotion as OOD, and filter these out from ChessBench—these include board states containing more pieces of a given type than normally allowed, e.g., 3 white queens, or positions with two bishops on squares with the same color. The original ChessBench was extracted from 10M games on Lichess (lichess.org an open-source online chess server), making the dataset size $\approx 528M$, from which we excluded $\approx 2.5M$. Only $0.43\%$ of the boards fell under what we define as OOD. After filtering, these situations have zero probability under the training distribution.

The board is represented by a FEN string (Edwards, 1994), which is a standard description of a chess position. It consists of a board state, the current side to move, the castling availability for both players, a potential en passant target, a half-move clock and a full-move counter, all represented in a single ASCII string. For example, the FEN of the starting position of the standard chess is `rnbqkbnr/pppppppp/8/8/8/8/PPPPPPPP/RNBQKBNR w KQkq - 0 1`, where the lowercase letters denote the black player, and the uppercase letters denote the white player. Actions are stored in UCI notation (Huber & Meyer-Kahlen, 2000), e.g., `e2e4` correspond to the popular opening where a white pawn moves from square `e2` to `e4`. The input of the Transformer is the tokenized FEN string and the output is the log probability distribution over all possible actions. Importantly, when generating the next move, we do not enforce it to be legal. Further training details are in § B.

### 3.2 OOD DATASETS

To assess the model's out-of-distribution (OOD) performance, we study 7 OOD datasets, which have zero probability under the training distribution. To confirm that the model operates in the regime of perfect legal move accuracy, we also evaluate it on 2 in-distribution (ID) sets.

**ID and OOD Puzzles.** The puzzle datasets are downloaded from Lichess[2]. These are curated in-distribution board states from games and corresponding puzzle solutions as sequences of moves . According to Lichess, all moves of the solution are "only moves", i.e., playing any other move would considerably worsen the player's position. Except for mate situations, where any move resulting in mate is correct. The puzzles having more pieces of a given type than allowed or having two bishops on same-colored squares form the OOD set, and the others form the ID set. Both datasets consist of 1000-1000 puzzles.

**ID test set, More pieces, and Same color.** The ChessBench dataset also includes a test set, which we filter in the same way as the training set: removing the aforementioned OOD boards. This results in 1000 ID board state–next move pairs. We separate the OOD boards into two datasets: the boards with more pieces of a given type than allowed forms *"More pieces"* (1000 boards; see Fig. 1, *first* board). Positions presenting two bishops on same-colored squares form the *"Same color"* dataset (1000 boards; see Fig. 1 *second* board). As these datasets where filtered from the test set, we will refer these datasets together as *"OOD test scenarios"*. Note that these datasets comprise boards from real games featuring pawn promotions making them more representative of late-game positions.

**Chess960 and All starting positions.** The next types of OOD boards feature starting positions in which the first-rank white pieces are randomly reordered, with the black pieces mirrored accordingly, but the pawns start in their standard position. Additionally, in *Chess960*, in the starting position, the king must be placed between the two rooks because of castling, and the two bishops must be on opposite-colored squares. In contrast, there are no such requirements for the *"All starting positions"* dataset, it contains all starting positions which cannot be reached from the classical starting position. There are 959 Chess960 starting positions without the standard starting positions (see Fig. 1, *third* board), and when evaluating the All starting positions dataset, 1000 boards are sampled from the possible starting positions.

**Chess variants.** We consider two chess variants to measure the model's ability to adapt to unseen scenarios. Chess960 is a popular, well studied variant (Gligoric, 2003; Deo & Dwivedi, 2023; Pav, 2025), where the game starts from the positions described above, but the goal remains the same. Standard chess involves extensive opening theory (Sterren, 2009), encouraging reliance on memorized sequences. Chess960 mitigates this by randomizing starting positions, requiring the model to depend on general principles and strategic reasoning. The second variant we study is Horde, which is a chess variant with White having 36 pawns (see Fig. 1, *fourth board*). The goal of White is to checkmate the black king, but as White does not have a king, Black wins by capturing all pieces of White. Note that the game is asymmetric, according to Lichess database, Black wins $52\%$ of the time, therefore it is the slightly more advantageous color.

**Knights&Rooks.** This dataset is a custom-made collection designed to explore the limits of OOD behavior. Each board contains a black and a white king, 2–4 white rooks to restrict the black king's movements, and 3–15 white knights. An example board, featuring 4 rooks and 15 knights, is shown in Fig. 2.

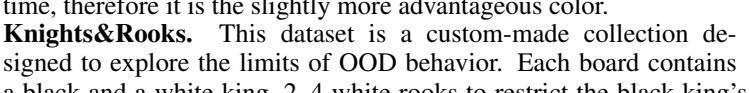

Figure 2: **Knights&Rooks:** extreme cases of boards having more pieces (knights and rooks) than normally allowed.

## 3.3 Evaluation

We evaluate the model on the ID and OOD datasets using the following metrics.

**Legal move accuracy** is calculated by the percentage of the model's moves that obey the rules of chess. For every board state, only the legality of the next move is evaluated.

**Stockfish topK accuracy** measures the quality of the model's predictions by determining whether the chosen move is among the topK move recommended by Stockfish. When multiple strong moves are available, expecting the Transformer's move to exactly match Stockfish's may not be representative. Therefore, we evaluate whether the chosen move falls within Stockfish's top3, top5, or top10 suggestions generated independently. We use Stockfish 17 with a search depth of 20, rather than a fixed time limit, to ensure a fair evaluation, since endgames and starting positions may require different amounts of time to produce moves of comparable quality. In § C.2, we perform extensive Stockfish ablations, examining the effects of search depth, comparing results with a fixed time limit, and evaluating different generating techniques for the top moves.

**Puzzle sequence accuracy** measures whether the model predicts correctly the entire move sequence of the solution—move sequences are $3.69 \pm 2.16$ long for the in-distribution, and $3.36 \pm 1.87$ for the OOD case on average. If any time during the sequence prediction a mate-in-1 situation occurs, every possible move that checkmates is considered correct. In this scenario only, the predicted move does not need to exactly match the move in the solution.

**Elo rating** is a standard metric of playing ability. We measure the model's Elo in three chess variants: Standard chess, Chess960, and Horde Chess—for descriptions of the variants, refer to § 3.2. We evaluate it in two settings: first, in an internal tournament, the model plays against 5 Stockfish engines with skill levels 0, 1, 2, 3, 4 out of 20. The skill level is weakened by introducing noise to the searching mechanism. According to Zhang et al. (2025), Stockfish level 0 has an Elo score of 1350-1440, level 1 has 1450-1560, and level 2 has 1570-1720 in Standard chess. Note that Stockfish, being a goal-driven symbolic AI, cannot play Horde, since Black's objective is not checkmate. Accordingly, for Horde we employ Fairy-Stockfish, which includes built-in support for chess variants. In the tournament, 100-100 games per opponent pair are played (altogether 500 per player), and every player plays half of their games as White and half of the games as Black. We compute the Elo score with relative BayesElo (Coulom, 2008) using the default confidence parameter of 0.5. The relative Elo score is used only to determine the playing strength order of the models, it cannot be directly compared to FIDE Elo ratings or to Elo ratings on Lichess. The average of the relative Elo scores of the models in a tournament is designed to be 0. To ensure variability of the games, for Standard chess, we use the openings from the Encyclopaedia of Chess Openings (Matanović, 1978), and for fair comparison, in Chess960, from the starting positions 10 full steps are made using the oracle Stockfish (depth 20, maximum skill level). We also made our model publicly available on Lichess, and let it play against both bots and humans. On Lichess the Glicko-2 system (Glickman, 2012) is used, as an improvement of the Elo system.

## 4 Results

To study to what extent our model understands chess, we distinguish between learning the rules and learning the strategy and study these in OOD scenarios—in models that achieve perfect in-distribution performance w.r.t. making legal moves. This is to disentangle whether the model overfit the data (which could follow from perfect ID performance) and to realistically assess OOD performance (if ID the model is far from perfect, we cannot have reasonable expectations OOD).

### 4.1 Rule extrapolation

Rule extrapolation is a term coined by Mészáros et al. (2024) for neural networks that can extract (language) rules during training and apply them in OOD scenarios. As chess is rule-based, we can apply the same investigative lens in OOD scenarios of different complexity (blue rows in Tab. 1). As a sanity check, we calculate the % of legal moves on in-distribution data (puzzles and the test set from ChessBench): the model gradually learns the rules of the pieces during training (Fig. 5 **right**), and the trained model achieved perfect score in these datasets. Even in OOD cases, our model almost always makes perfect moves—apart from the Knights&Rooks scenario, it achieves $96 + \%$. **Our model also makes mostly legal moves on the Knights&Rooks, which is designed to explore the limits of OOD behaviour. Surprisingly, it is also able to play Chess960 and Horde**, by making legal moves in $99.36\%$ and $95.96\%$ of the time, respectively (see Tab. 2). When the model played

| Dataset | Accuracy (%) | | | | | |
|---|---|---|---|---|---|---|
| | Legal | Sf. top1 | Sf. top3 | Sf. top5 | Sf. top10 | Puzzle seq. |
| ID Puzzles | 100 | 70.50 | 87.17 | 92.56 | 96.88 | 58.80 |
|    Test set | 100 | 56.30 | 79.48 | 86.62 | 94.24 | - |
| OOD Puzzles | 99.60 | 67.70 | 84.81 | 89.04 | 92.93 | 54.70 |
|    More pieces | 97.20 | 30.49 | 39.53 | 37.12 | 43.12 | - |
|    Same color | 97.60 | 30.60 | 45.54 | 45.18 | 50.46 | - |
|    Chess960 starting pos. | 96.45 | 22.73 | 52.24 | 66.42 | 88.80 | - |
|    All Starting pos. | 97.00 | 22.80 | 49.90 | 66.00 | 84.60 | - |
|    Knights&Rooks | 90.20 | 2.00 | 3.70 | 6.30 | 13.80 | - |

Table 1: **Rule extrapolation ("Legal" col.) and Strategy adaptation (other cols.) accuracies over the ID and OOD datasets:** the model has perfect legal next move accuracy on the ID datasets, and almost always makes legal moves on the OOD sets, even on the highly OOD Knights&Rooks. In terms of strategy adaptation, Sf. top1 in OOD Puzzles is only marginally worse than in the ID case, and is still non-trivially large on the other sets. Also, there is a clear inverse relationship between the number of possible good moves and the Sf. top1 accuracy. The model could not adapt to the extremely OOD nature of the Knights&Rooks dataset. For details, refer to § 4

Horde games in Black, the legal move accuracy was slightly higher ($97.18\%$) compared to when it played Black ($94.74\%$).

Sometimes move illegality arises not from violating a piece's movement rules (i.e., the model is not trying to move a bishop, e.g., vertically) but by moving pinned pieces (when a piece is not allowed to move because that puts the king into check). However, this only occurs rarely—from 342 cases when there is a pinned piece on board, it only fails in 4 cases (Fig. 3). Notably, these are the *only* cases when the model does not predict a legal move for the OOD Puzzles.

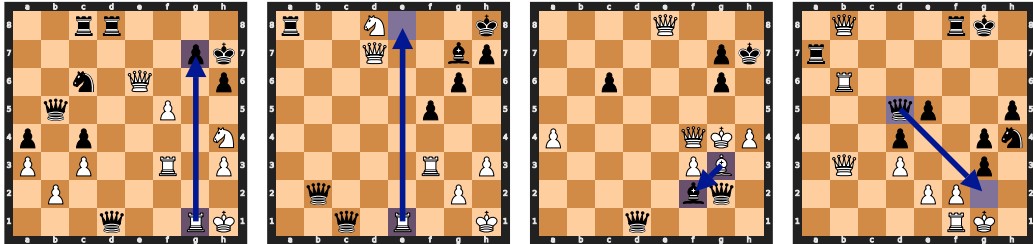

Figure 3: **The illegal moves on OOD puzzles** arise from the model trying to move pinned pieces. E.g. on the *fourth* board, moving the black queen `d5g2` would mate the white king, but it allows to the white queen on `b3` to put the black king into check.

## 4.2 STRATEGY ADAPTATION

Strategy adaptation goes beyond rule extrapolation as it entails not just knowing the correct rules, but making the best next move, constrained by the game's rules. Measuring the optimality of a move is less obvious than checking move legality. We can evaluate against the Stockfish engine, though that is not entirely deterministic[3]. Despite this fact, it is a good indicator of move quality, as used in previous works (Ruoss et al., 2024; Feng et al., 2023; Zhang et al., 2025). When there are multiple potential good next moves, predicting the same one as Stockfish might not be perfectly indicative of whether a model makes a good move. Therefore, we measure not just the top1 accuracy, but also top3, top5 and top10 accuracies. However, boards are not guaranteed to have K next steps. In that case, we average over a different number of boards—for details refer to § C.1. Generating the moves independently makes it possible that there is a slight decrease in accuracy even for a larger K value in More pieces and Same color.

---

[3]One reason for this is the neural network–based NNUE module

**OOD datasets.** We summarize our findings in the following and detail them below:

1. For the **puzzles**, OOD Stockfish accuracy is very close to the ID one, with the top3 accuracy already being above 80%, the top5 being very close to 90% and top10 is above 90%.
2. For the **OOD test scenarios** that are reachable in standard chess through pawn promotion (More pieces, Same color), the Stockfish topK accuracy is significantly lower, though still non-trivially large (above 30%).
3. For the **starting positions**, the Stockfish top1 accuracy is the lowest ($\approx 23\%$), but the top10 accuracy increases to $84 - 88\%$.

An important distinction between the puzzles and OOD test scenarios is that the puzzles were *"supervised"* in the following sense: According to Lichess, the player moves for the puzzles are "only moves", i.e., playing any other move would considerably worsen the player's position. This is not true for the other datasets. Thus, it is easier for our model to predict the same best next move as that of Stockfish for the Puzzles.

Another difference is whether the dataset consists of early-game (All starting pos., Chess960 starting pos.), more late-game (More pieces, Same color), or end-game (Puzzles) positions. In this order, the number of potentially good moves decreases rapidly (in starting positions, there are several good openings to play, but for puzzles, often there is only a single good move). This is reflected in Tab. 1 Stockfish accuracies, especially in the *Sf. top1* columns. As the puzzles, the More pieces and Same color data sets capture late-game boards, the amount of good moves is smaller; thus, it is easier for the model to predict the same move as the Sf engine. Also, late-game bards require a shorter planning horizon. The only failure case for strategy adaptation is the Knights&Rooks scenario, which was designed to explore the limits of OOD behavior. The model could not adapt to the highly OOD nature of the boards, which were extremely divergent from the training data.

Chess960 and All starting pos. posits a seemingly surprising dichotomy: while the top1 and top3 Sf accuracies are among the lowest, the top5 and top10 are among the highest (except the puzzles). The reason behind this is the early vs late game differences across the evaluation scenarios. Namely, the number of possible legal moves is very limited in starting positions (only pawns and knights can move). or example, for starting positions where the pieces on the 1st rank are randomly reordered, there at most 20 legal moves. However, neither Stockfish nor our model moves a pawn by only one square, leaving only 12 legal moves. For details on the number of possible starting scenarios, see § D. As Stockfish only predicts legal moves, if we chose K large enough, the moves predicted by the Transformer will be among them with very high probability (since the Transformer also mostly predicts legal moves). As the number of possible legal moves is much larger in the late-game scenarios More pieces and Same color, it is necessary that for a large enough K, accuracy for these two scenarios will be lower. This also implies that as K increases, the increase in late game accuracies will be lower than for Chess960 and All starting pos., meaning that even if these have higher top1 accuracy than More pieces and Same color, the relationship will flip—empirically, it already flips for our model for K=3.

**Tournaments.** In terms of the Puzzle sequence accuracy, the model achieves $58.80\%$ on the ID dataset, which is not surprisingly lower than the Stockfish top1 accuracy as the model have to predict not one but a sequence of moves. On the OOD dataset, the accuracy is only slightly lower ($54.70\%$), indicating that the model is able to adapt comparab to the OOD situations and show non-trivial performance.

| Variant | Legal acc. % | Lichess Elo |
|---|---|---|
| Standard | 99.92 | $1550_{\pm 45}$ |
| Chess960 | 99.36 | $1571_{\pm 51}$ |
| Horde | 95.96 | $1178_{\pm 68}$ |
| HordeW | 94.74 | - |
| HordeB | 97.18 | - |

Table 2: **Legal % and Lichess Elo:** Legal move accuracy in tournaments and Lichess Elo of our model across variants. HordeW/HordeB show results when our model played White/Black.

The tournament setting measures long-horizon planning and reasoning in our model, as it needs to play whole games against different Stockfish configurations (for details, refer to § 3.3)—or bots/humans on Lichess. Note that even in the standard chess tournaments, our model can face situations that are OOD regarding our training set—namely, pawn promotion can occur, and it does occur, making $0.29\%$ of the boards OOD when it is the model's turn. This potentially explains the $99.92\%$ not perfect Legal move accuracy on Standard chess (Tab. 2).

In the tournaments not conducted via Lichess, we used three different chess variants. In the order of increasing OOD complexity, these are: standard chess, Chess960, and Horde chess. **In standard chess, our model places** 3rd, **though in terms of relative Elo, it comes very close to the second-**

| Standard | Rel. Elo | Draws | Chess960 | Rel. Elo | Draws | Horde | Rel. Elo | Draws |
|---|---|---|---|---|---|---|---|---|
| 1. Sf.4 | $205_{\pm 28}$ | 3% | 1. Sf.4 | $240_{\pm 30}$ | 2% | 1. FSf.4 | $384_{\pm 38}$ | 0% |
| 2. Sf.3 | $114_{\pm 26}$ | 2% | 2. Sf.3 | $169_{\pm 27}$ | 3% | 2. FSf.3 | $239_{\pm 32}$ | 0% |
| 3. **Trf** | $\mathbf{88_{\pm 26}}$ | **5%** | 3. Sf.2 | $14_{\pm 26}$ | 3% | 3. FSf.2 | $10_{\pm 29}$ | 0% |
| 4. Sf.2 | $-27_{\pm 26}$ | 3% | 4. Sf.1 | $-86_{\pm 26}$ | 3% | 4. FSf.1 | $-61_{\pm 29}$ | 0% |
| 5. Sf.1 | $-126_{\pm 27}$ | 1% | 5. **Trf** | $\mathbf{-110_{\pm 26}}$ | **5%** | 5. FSf.0 | $-223_{\pm 31}$ | 0% |
| 6. Sf.0 | $-253_{\pm 31}$ | 1% | 6. Sf.0 | $-227_{\pm 29}$ | 1% | 6. **Trf** | $\mathbf{-350_{\pm 36}}$ | **0%** |

Table 3: **Tournament results:** The figure shows the tournament results of the model (**Trf**) against Stockfishes level 0-4 in Standard Chess (**Left**) and Chess960 (**Middle**) and against Fairy-Stockfishes level 0-4 in Horde Chess (**Right**). Our model places 3rd in Standard chess, 5th in Chess960, and 6th in Horde. The % of draws of the games the players had is also reported.

**placed Stockfish level 3. In Chess960, it places 5th with a close gap to the 4th Stockfish level 1 engine. The Transformer clearly ranks last in the Horde chess tournament.**

Stockfish, being a goal-driven engine, cannot play Horde, since Black's objective is not checkmate, so for Horde we use Fairy-Stockfish. Surprisingly, our model can play Horde legally and even win against Fairy Stockfish: it won 31 games out of 250 when playing white, 76 out of 250 when playing black. We want to emphasize that Horde chess represents a significant distribution shift, as our model either needs to adapt to having only pawns (when playing White) or playing against an opponent without a king (when playing Black). When playing Black, the model also needs to optimize for a different objective: instead of mating a (non-existent) white king, it needs to capture all white pawns. Indeed, when playing Black, **our model learns the strategy that capturing pieces is advantageous**. Moreover, it even performs better when playing with pieces from the classical starting position but with different aim, than with 36 pawns but the original goal of chess. However, it still underperforms all Stockfish variants in our tournament (Tab. 4).

On Lichess, our model was playing against both humans and bots and reaches a Lichess Elo score of 1550 in Standard Bullet chess (better than 48.5% of players), of 1571 in Chess960 (better than 42.9% of players), and of 1178 in Horde (better than only 8.4% of the players) (Tab. 2). **We conclude, that the model's Chess960 playing ability is almost as good as the Standard chess playing ability against players on Lichess, but fails to the strategies of Horde.** Note that we cannot control who plays against our model, therefore we report the deviation in the ratings Tab. 3, the % of the games played by human and the average Elo of the opponents. More details and statistics about the games played on Lichess, the % of the games played by human and the average Elo of the opponents and the average Elo of the opponents can be found in § E.1.

| White \ Black | FSf.4 | FSf.3 | FSf.2 | FSf.1 | FSf.0 | **Trf.** |
|---|---|---|---|---|---|---|
| FSf.4 | - | 26 | 41 | 44 | 49 | 48 |
| FSf.3 | 13 | - | 30 | 34 | 42 | 48 |
| FSf.2 | 2 | 6 | - | 19 | 34 | 39 |
| FSf.1 | 1 | 3 | 11 | - | 21 | 41 |
| FSf.0 | 1 | 2 | 2 | 6 | - | 29 |
| **Trf** | 2 | 2 | 10 | 6 | 11 | - |

Table 4: **Win count in Horde** The figure shows the number of games won by the models in White against the models in Black. The models include Fairy-Stockfish level 0-4 and our Transformer model. In each cell, the total number of played games is 50. Our model won 31 games as White and 76 as Black.

While the models fails to play Horde well both in the tournament and on Lichess, one can observe that when the model plays against humans/bots, its playing ability of Chess960 is close to Standard chess Tab. 2. However, the difference seems larger when playing in tournament against Stockfish. We hypothesize that this happens because Stockfish plays the two variants in the same way (expicitly searching for the best move), but humans rely much more on statistical patterns and memorized openings.

Furthermore, the model is easily drawn by threefold repetition (when the same board occurs three times), because the model does not keep track of the past moves of the game, only sees the current board. Stockfish can detect possible threefold repetition, making the % of draws small in the tournament Tab. 3. However, when the model plays against humans (and bots) on Lichess, the draw % is much higher Tab. 9.

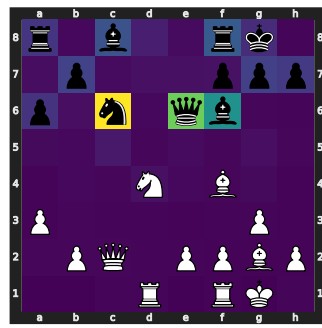 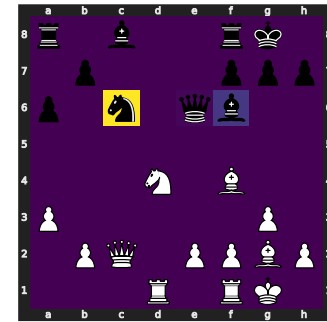

Figure 4: **Training dynamics of piece selection**: Heatmap of an ID Puzzle board showing summed probabilities of next moves starting from each square, depicted at initialization **(left)**, during training **(middle)**, and after training **(right)**. On each board, the probabilities are normalized to $[0; 1]$ by the maximum value, dark blue indicates 0, yellow indicates 1

### 4.3 TRAINING DYNAMICS

We study the dynamics of how our Transformer model learns to select the piece for its next move. We illustrate this process with an in-distribution puzzle (Fig. 4, showing initialization, mid-training, and end-of-training probabilities from left to right). From an approximately uniform probability distribution at initialization, **the model first learns to move with its own (black) pieces** (Fig. 4, middle) shown by the probabilities concentrating on black pieces. It also learns that the pawn on f7 is not movable. At the end, it picks the black knight by concentrating the probability mass to it.

We also investigate the dynamics of learning all legal moves, both ID and OOD (Fig. 5). The model first learns to generate a single legal move on the ID boards, then on the OOD boards. After that, it starts to identify all legal moves of the ID positions and then on the OOD positions by assigning them higher probability (see Fig. 5 **middle**). At the end of the training, all curves reach almost perfect legal move accuracy. Note that the majority of the change in the probabilities occurs at the beginning of the training (unit 1M steps), followed by a slower convergence to 1 (Fig. 5 **left**).

## 5 CONCLUSION

In this work, we investigated the extent to which Transformer-based chess policies display systematic generalization beyond their training distribution. Our experiments demonstrate that the model exhibits compositional generalization, as evidenced by strong *rule extrapolation*: it reliably adheres to the syntactic rules of chess even in novel and highly out-of-distribution positions. This capacity enables it to play valid and often strategically sound moves in puzzles and variants very different from its training data. When tested on challenging variants such as Chess960 and Horde, the model shows partial but limited *strategy adaptation*, highlighting the gap between implicit generalization in black-box neural policies and explicit compositional reasoning in search-based symbolic algorithms. By contrast, the gap is much smaller against players on Lichess, where the model's Chess960 playing ability is nearly on par with its Standard chess playing ability. The training dynamics revealed that the model initially learns to move only its own pieces, suggesting an emergent compositional understanding of the game. Nevertheless, the fact that a purely behaviour-cloned Transformer can generalize to legal and strategically plausible play across diverse out-of-distribution settings indicates that these models capture more compositional structure than would be expected from mere statistical pattern matching.

**Limitations** Our chess Transformer shows promising signs of compositional generalization by extrapolating the rules to substantially different OOD scenarios. However, the model's strategic adaptation remains limited: while it reliably follows the rules of chess, it struggles in scenarios requiring long-term planning or novel strategies, such as the Horde variant or high-level play against Stockfish. Also, we could not control who plays against our model on Lichess, which may introduce bias to the rating.

## REPRODUCIBILITY STATEMENT

To ensure reproducibility of our results, we will release the full codebase, trained model checkpoints, and the datasets used in this study upon acceptance of the paper.

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

## A   TRAINING DYNAMICS

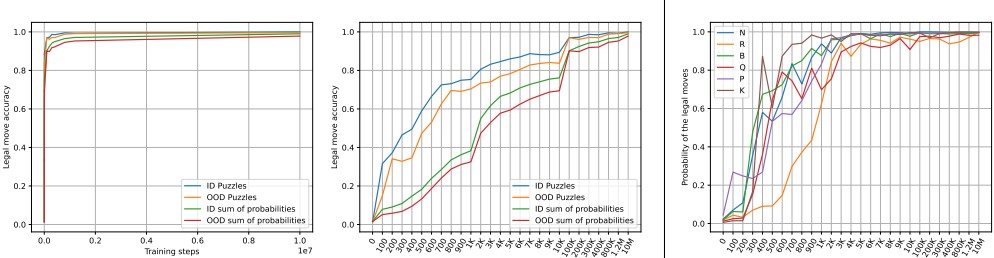

Figure 5: **Training dynamics of move legality**: the **left** and **middle** plots illustrate the legal *next* move accuracy during training on the ID (blue) and the OOD Puzzles (orange); and the sum of all legal moves' probabilities from all possible moves for the ID (green) and the OOD Puzzles (red). Averages are taken over 1000 puzzles. The two plots are scaled differently to better see the beginning of the training. The **right** plot shows the relative legal probability of the pieces, calculated as $p(\text{legal moves of a given piece})/p(\text{all moves with a given piece})$. The notation of the pieces is the following: N - knight, R - rook, B - bishop, Q - queen, P - pawn, K - king. These values were calculated on very simple boards containing one black, one white king and only the type of piece whose legal move was examined.

## B   EXPERIMENTAL DETAILS

Our Transformer model uses the same setup as Ruoss et al. (2024), excluding the dataset and the bach size. Our the exact size of our filtered dataset is $525\,388\,668$ as described in § 3.1. For the batch size, we choose 256, which is more suitable to our GPU. The experimental details can be found in their paper. Here we only detail some hyperparameters. A decoder only Transformer is trained with 16 layers, 8 heads and embedding dim of 1024, making the model $\approx 270M$ parameters. The context length was 78 from which 77 corresponds to the tokenized FEN, and one for the next action. When the model is used for prediction, a dummy action character appended after the FEN. THE output dim is 1968, corresponding to all the possible actions. The model is trained for 10M steps, which corresponds to 4.87 epochs. For prediction, the `argmax` of the output probabilities is chosen.

To train the model, we used 4XH100 GPUs (80GB SXM5) and the evaluation is run on a dual-socket Intel(R) Xeon(R) Gold 6526Y CPU (32 physical cores, 64 threads, up to 3.9 GHz) and four NVIDIA L40S GPUs (46 GB memory, CUDA 13.0).

## C   STOCKFISH

### C.1   DETAILS OF TAB. 1

As it was described in § 4.2, when there are no K legal moves from a board, Stockfish cannot generate topK best moves. In Tab. 5, we report the number of boards top1,3,5,10 were calculated on regarding each dataset.

### C.2   STOCKFISH ABLATIONS

In this section we describe the extensive Stockfish ablations we conducted. This was necessary, because the Stockfish engine used for generating the test labels (see § 3.1) cannot be reproduced as the quality of the engine with 0.05 time limit per move greatly depends on the chip Ruoss et al. (2024) used for running it. Also, they used an older version of Stockfish (16), but the time we are writing the paper a newer version (17) is available. Therefore, we calibrate Stockfish 17. Fist, we compare Stockfish with different time and depth limits to ground truth values. We use depth as the limit, rather than a fixed time, to ensure a fair evaluation, since endgames and starting positions may require different amounts of time to produce moves of comparable quality. From Tab. 6, we conclude that depth 20 only marginally worse than depth 30, but requires much less compute time.

| Dataset | Accuracy (%) | | | |
|---|---|---|---|---|
| | Sf.Top1 | Sf.Top3 | Sf.Top5 | Sf.Top10 |
| ID Puzzles | 1000 | 990 | 968 | 898 |
| ID test set | 1000 | 965 | 934 | 886 |
| OOD Puzzles | 1000 | 981 | 958 | 947 |
| More pieces | 1000 | 819 | 653 | 543 |
| Same color | 1000 | 774 | 625 | 545 |
| Chess960 starting pos. | 959 | 959 | 959 | 959 |
| All starting pos | 1000 | 1000 | 1000 | 1000 |
| Knights&Rooks | 1000 | 1000 | 1000 | 1000 |

Table 5: **Ablation on the method of choosing the topK move of Stockfish:** On the More pieces dataset (1000 boards), we evaluate Stockfish with varying depth and method of choosing topK, and report the top1,3,5,10 accuracies. Moreover, in (parentheses) we show the number of boards the accuracies were calculated on.

| | Accuracy (%) | | | | Accuracy (%) | |
|---|---|---|---|---|---|---|
| Time limit | Puzzles | Test label | | Depth limit | OOD Puzzles | Filtered test set |
| 0.05 | 98.80 | 61 | | 10 | 97.20 | 52 |
| 0.5 | 99.20 | 62 | | 15 | 98.80 | 62 |
| 1 | 99.20 | 63 | | 20 | 99.20 | 63 |
| 1.5 | 99.40 | 63 | | 30 | 99.40 | 64 |

Table 6: **Stockfish with varying limits compared to ground truth values** Table on the **left:** Stockfish is evaluated on 1000 OOD Puzzles whether the next move predicted by the engine equals the *first* move of the solution of the puzzle. While on the **right:** On 100 boards from the (ID) filtered test set, we measure whether the move of Stockfish is the same as the test label.

Even though we do not know the exact Elo scores of the engine, but based on Ruoss et al. (2024), we hypothesize that a depth 20 Stockfish 17 has 3000 Elo score making it a very strong engine. For comparison, the world no.1 chess player, Magnus Carlsen, has 2839 Elo in classical chess at the moment (September 2025).

Next, we compare Stockfish to our Transformer model (Tab. 7).

As it can be seen in Tab. 1, we evaluate whether the Transformer's next move is among the topK (K=1,3,5,10) moves of Stockfish. For this, an ablation is made on how we choose the topK moves. We compare the case where top1,3,5,10 are independently generated to the case where top10 is generated but top1,3,5 are chosen as the top moves among top10 (Tab. 8). These methods have different outcomes.

Generating only top10 (then choosing top1,3,5 from it) seems unfair, because there are only 545 cases when Stockfish can generate 10 moves, therefore top1,3,5 will be estimated on 545 boards, too. Thus, all the boards, where there are no 10 different moves (very endgame positions) will be left out. Intuitively, generating moves for endgame positions is easier, so leaving them out would unfairly lower the accuracies. In fact, generating top1 in-

| | Accuracy (%) | |
|---|---|---|
| Depth | OOD Puzzles | Filtered test set |
| 10 | 67.90 | 42 |
| 15 | 67.70 | 52 |
| 20 | 67.70 | 57 |
| 30 | 67.00 | 53 |

Table 7: **Stockfish compared to the Transformer model:** On 1000 OOD Puzzles and on 100 positions from the filtered test set, we evaluated whether the Transformer's next move is equal to Stockfish's next move.

dependently with even depth 20 significantly increases accuracy compared to the case when top1 in chosen from top10 with depth 30.

Consequently, we choose to generate top1,3,5,10 moves independently. The only downside is that it can lead to non-consistently increasing values (top3 ¿ top5). One can argue that it should be allowed

more searching for generating top10 than top1, but allowing searching depth 30 does not really differ from depth 20.

| Method | Accuracy (%) | | | |
|---|---|---|---|---|
| | Sf.Top1 | Sf.Top3 | Sf.Top5 | Sf.Top10 |
| Depth 20 + generate all top1,3,5,10 | 30.40 (1000) | 39.53 (774) | 37.12 (625) | 43.12 (545) |
| Depth 30 + generate top10 and choose top1,3,5 | 13.30 (545) | 26.24 (545) | 31.01 (545) | 43.85 (545) |
| Depth 20 + generate top1,10, and choose top3,5 | 30.40 (1000) | 24.95 (545) | 31.01 (545) | 43.12 (545) |
| Depth 30 + generate top1,3,5,10 | 29.90 (1000) | 41.09 (774) | 38.88 (625) | 43.85 (545) |

Table 8: **Ablation on the method of choosing the topK move of Stockfish:** On the More pieces dataset (1000 boards), we evaluate Stockfish with varying depth and method of choosing topK, and report the top1,3,5,10 accuracies. Moreover, in (parentheses) we show the number of boards the accuracies were calculated on.

## D  OPENING MOVES OF STARTING POSITIONS

From a starting position described in § 3.2, there can be at most 20 legal moves, as the first piece to move is either a pawn or a knight. For example, on the board Fig. 6, each of the 8 pawns can move 1 or 2 squares forward, and each knight can be placed to 2 squares: the knight on `c1` can move to `b3` or `d3`, and the knight on `d1` to `c3` or `e3`. If we assume that most of the time when a player moves a pawn as a very first move of the game, the pawn is moved by 2 squares, then the effective number of legal moves is 12.

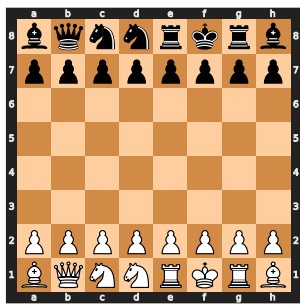

Figure 6: **Starting position:** a starting position from the Chess960 dataset.

The model tends to play the same openings for Chess960 starting boards, `d2d4`, `c2c4`, `e2e4`, `f3f4` as the knight moves cover ≈ 90% of the cases. The model plays `d2d4` 360, `c2c4` 335, `e2e4` 70, `f2f4` 44, `c2c3` 10, `a2a4` 8, `b2b4` 7, `f2f3` 7, `h2h4` 4, `d2d3` 3, `g2g3` 2, `g2g4` 1, `a2a3` 0, `b2b3` 0, `e2e3` 0, `h2h3` 0 times, and moves the knight 74 times. While Stockfish generates `d2d4` 150, `e2e4` 148, `f2f4` 124, `c2c4` 117, `b2b4` 89, `a2a4` 80, `g2g4` 79, `h2h4` 61, `g2g3` 22, `c2c3` 19, `b2b3` 18, `f2f3` 12, `e2e3` 8, `d2d3` 6, `a2a3` 0, `h2h3` 0 times, and moves the knight 26 times. Here, the top10 moves cover the 92.7% of the cases.

## E  ELO RATINGS

### E.1  LICHESS

In this section, the details of the games played on Lichess can be found. In Standard chess, the model played Blitz games, too, in which it achieved an Elo score of $1493_{\pm 61}$ making it better that the 51% of the players on Lichess. The number of games the Elo ratings are based on is 100 for Standard Bullet, 100 for Chess960, 50 for Horde, and 55 for Standard Blitz. This is the reason why the deviation of the rating is higher in the case of Horde and Blitz. The win/draw/loss percentages can be seen in Tab. 9. The average Elo of the opponent was 1572 in Standard Bullet, 1544 in Blitz, 2041 in Chess960 and 1650 in Horde. The 14% of the Standard Bullet, 13% of Standard Blitz, 57% of Chess960 and 76% of Horde games was played by a human.

|      | Bullet | Blitz | Chess960 | Horde |
|------|--------|-------|----------|-------|
| win  | 24%    | 35%   | 23%      | 14%   |
| draw | 41%    | 35%   | 14%      | 32%   |
| loss | 35%    | 30%   | 63%      | 54%   |

Table 9: **Draw** percentages of the games playen on Lichess.

### E.2  TOURNAMENTS

In addition to the realtive Elo in the Tournaments, we report the score of each player, which defined as

$$\text{score} = \frac{1 * \#\text{wins} + 0.5 * \#\text{draws}}{\#\text{all games}}.$$

The scores can be found in Tab. 10. The order of the models based on their score is identical to the order based on their relative Elo ratings in every chess variant.

| Model | Standard | Chess960 | Horde |
|-------|----------|----------|-------|
| **Trf** | **61%** | **36%** | **15%** |
| Sf.4 | 75% | 79% | 88% |
| Sf.3 | 64% | 71% | 76% |
| Sf.2. | 46% | 52% | 51% |
| Sf.1 | 34% | 39% | 44% |
| Sf.0 | 19% | 23% | 27% |

Table 10: **Scores:** of the models played in the tournament of the different variants.

