# OpenReview forum: "Out-of-distributon Tests Reveal Compositionality in Chess Transformers"
_ICLR.cc/2026/Conference — ICLR 2026 Conference Withdrawn Submission_

### Official Review · Reviewer_koTU · 2025-10-30

**Soundness:** 3
**Presentation:** 2
**Contribution:** 2
**Rating:** 4
**Confidence:** 4

**Summary:**

The authors introduce a number of OOD chess test sets based on variations of the game (Horde) and rare game positions (multiple pieces of the same color.) They provide baseline results over these tasks with a model trained on a filtered version of ChessBench (to help ensure these OOD tasks are in fact, OOD.)

**Strengths:**

* The work presents a number of cool ideas and environments for studying a well-known reasoning domain (chess) but in a natural, OOD-manner.

**Weaknesses:**

* Hard to know exactly what to takeaway. For most of the OOD tests the models are doing much worse, 20-30% vs 70%. Are the success cases examples of generalization or scenarios where the model can mostly ignore the "novel" parts? For example, with same-color, perhaps the model some rate of the time is not really generalizing in a "good" way but the examples are close enough to where the same-color interference is not that important?
* There might be a difference between generalizing--using what was learned and applying it to new situations, using the same concepts in new ways--versus *making do* or *coping*, using what is learned on the familiar parts of a new situation and behaving poorly/randomly/worse on the novel parts. In any case, it is unclear what partial generalization means in this paper, or what to take away from it.
* Overall I found the learning dynamics section unconvincing. It largely just shows one qualitative example?
* You spend two long paragraphs discussing the intricacies of how number of moves available and top# accuracy interact (in perhaps unexpected ways). However, this is not really capturing something useful about the models, but rather an artifact of the task. Answering some of the questions above (or some other analysis) would help. Some further insight or qualitative analysis would help strengthen the analysis of this section, and this section seems to be the crux of this work.

**Questions:**

* Q: Is there a reasonable way to weight the top-K moves from stockfish? Or dynamically determine a threshold for K? This largely is covered by the direct top-K approach but when there are limited numbers of moves or in some cases big quality differences among the top-k moves something like this could be helpful.
* Q: What kinds of questions are the models getting right? What kinds of questions are the models getting wrong?

---

### Official Review · Reviewer_vt1Y · 2025-10-30

**Soundness:** 3
**Presentation:** 2
**Contribution:** 1
**Rating:** 2
**Confidence:** 5

**Summary:**

This paper investigates to which extent Transformer models trained on chess exhibit systematic generalization and compositional understanding. The authors train a 270M parameter Transformer on a large dataset of chess games to predict the next best move (behavior cloning from Stockfish). They then test this model on a set of OOD scenarios. The authors frame their investigation around two key concepts: Rule Extrapolation (adhering to the game's rules in novel situations) and Strategy Adaptation (adjusting its plan to win in unfamiliar settings).

The key findings are:
- The Transformer exhibits strong rule extrapolation, making legal moves with high accuracy (>96%) even in unseen board states.
- The model shows partial but limited strategy adaptation. It can find high-quality moves in OOD puzzles but struggles in chess variants that require fundamentally new strategies, like Chess960 (randomized start) and especially Horde (asymmetric goals and pieces).
- Despite poor performance against optimized symbolic engines (Stockfish) in variants, the model's performance gap is smaller against human players on Lichess, suggesting it has learned human-like strategies.
- Analysis of the training dynamics reveals an emergent understanding, where the model first learns to move its own pieces before learning the full rules of the game.

Overall, the paper provides evidence that Transformers can learn the "syntactic" rules of a complex domain like chess in a compositional way, but struggle to dynamically reason and adapt their "semantic" strategy.

**Strengths:**

- The study is methodologically sound.
- The authors are careful to create a reliable OOD evaluation.
- Considering strategic adaptation is new and is valuable from the generalization perspective.

**Weaknesses:**

- The paper does not bring enough novelty compared to https://arxiv.org/abs/2412.12119. The two papers are *very* similar, yet the other paper is not cited.

- The contributions listed are not novel enough. Below I copy pasted the contributions:
• we created a battery of chess puzzles that present out-of-distribution situations (1)
• we trained a model-free Transformer-based chess policy using behaviour cloning reproducing the
methodology of Ruoss et al. (2024), but using a training dataset more suitable for our study; (2)
• we evaluated the models on rule extrapolation and strategy adaptation; (3)
• we evaluated the policy’s ability to play full games in Chess960 (Fisher Random Chess), in which the starting positions of pieces are randomized, and Horde, where the Black’s objective is altered; (4)
• we analyzed the dynamics of rule learning. (5)
- Contribution (1) would be valuable if the code is released, but does not bring much novelty.
- (2) and (4) are done in https://arxiv.org/abs/2412.12119 (the latter does not evaluate on Horde though)
- (3): I listed this in the strengths.
- (5) is interesting but lacks generality: the reported results are qualitatively worse than https://arxiv.org/abs/2412.12119, highlighting that the conclusions they draw might not hold with a stronger model.

- The paper is very dense, there is not enough space between the sections.

**Questions:**

- What new knowledge does your paper bring that would be transferable across different model sizes or trainings?
- Could you add https://arxiv.org/abs/2412.12119 to the references and add a discussion about the differences between these two papers?

---

### Official Review · Reviewer_r9CB · 2025-11-03

**Soundness:** 3
**Presentation:** 3
**Contribution:** 1
**Rating:** 2
**Confidence:** 5

**Summary:**

The paper investigates compositional generalization of chess transformers. Concretely, it trains a 270M-parameter transformer via behavioral cloning on (roughly) the chess benchmark from Ruoss et al. (2024). The paper then constructs 7 different OOD evaluation datasets to measure the model’s legal move accuracy and the quality of its move predictions (by comparing it to Stockfish’s predicted moves). The paper finds that the model has learned to predict legal moves in a large variety of OOD settings, including Chess960 and Horde. The paper also finds that the model’s playing strength is roughly similar for standard and Fischer Random chess, but quite a bit worse on Horde when evaluated on Lichess.

**Strengths:**

The paper conducts an additional evaluation of a previously proposed approach (Ruoss et al., 2024) and presents a comprehensive set of OOD datasets. The finding that transformers trained on standard gameplay achieve high legal move accuracy on Chess960 and Horde is both surprising and interesting to the community. The paper is reasonably well written and generally easy to follow. The paper adequately discusses related work.

**Weaknesses:**

It is not quite clear to me what the paper actually contributes over Ruoss et al. (2024). The model trained in this work is essentially equivalent to the one trained by Ruoss et al. (2024), modulo a very small change in the dataset (which this paper made to facilitate some OOD evaluations). However, Ruoss et al. (2024) _already_ computed the legal move accuracy, albeit only for standard chess. Ruoss et al. (2024) also _already_ evaluated puzzle-solving accuracy. Moreover, Ruoss et al. (2024) _already_ evaluated the Elo for standard chess and Chess960 in essentially the same way as this paper. So the only additional contributions I can make out are: (i) evaluating the legal move accuracy for Chess960, Horde, and some OOD datasets, (ii) evaluating the model’s playing strength on Horde. It is also not quite clear to me what the paper's results imply beyond chess transformers.

The paper trained a model similar to Ruoss et al. (2024), using essentially the same dataset. However, the paper’s model is quite a bit worse than the model trained by Ruoss et. al (2024), which achieved a Lichess Elo of 2299 against bots, whereas the model presented here only achieved an Elo of 1550. The paper does not explain this discrepancy.

The related works section is a bit long-winded and discusses work that is at most tangentially related (e.g., RASP).

There are quite a few typos and mistakes. I won’t list all of them here, but see, e.g.,
* “~the~ we use” (L146)
* “Both datasets consist of 1000-1000 puzzles” (L182)
* “When the model played Horde games in Black [...] compared to when it play Black.” (L291)
* “or example” (L350)
* “comparab” (L366)

**Questions:**

* In the introduction, the paper states that Stockfish is “highly interpretable”. Stockfish consists of 16K lines of code and employs a neural network-based evaluation function. Is this really “highly interpretable”?
* The paper states that “when playing black, our model learns the strategy that capturing pieces is advantageous.” How does the model learn this _at evaluation time_?

---

### Author Response · Authors · 2025-11-20
**Response to Reviewer Comments: Clarifications Regarding Novelty and Contribution**

We sincerely thank the reviewers for their thoughtful comments and for acknowledging the comprehensiveness of our out-of-distribution (OOD) evaluation, the novelty of the strategy adaptation perspective, and the interest of several of our empirical findings.

The main concern raised relates to the novelty of our contribution in light of two recent papers (Ruoss et al., 2024; Schultz et al., 2025). We appreciate the suggestion to discuss these works more thoroughly, and we clarify below how our study is complementary to, and distinct from, these approaches.

1. **Distinct research objectives**
Ruoss et al. and Schultz et al. focus primarily on improving chess-playing performance, employing techniques such as legal-move constraints, external or internal search, or training on specific chess variants. Their aim is to build stronger chess models.
In contrast, our paper is an empirical study of transformer reasoning and OOD generalization. Chess serves only as a controlled domain for analysis, not as a target task for performance optimization.
2. **Broader and more systematic OOD evaluation**
Ruoss et al. evaluate accuracy only on in-distribution datasets. Schultz et al. include an OOD-like setting involving random boards, but the evaluation is limited to a single dataset and focuses on legal-move accuracy.
Our work provides a substantially broader OOD evaluation across multiple dataset types, introduces additional metrics and stress tests, and uncovers several behavioral patterns not reported in prior work.
3. **Novel conceptual contributions**
Our analysis of strategic adaptation and training dynamics introduces perspectives not explored in the cited papers. Reviewers correctly noted that both papers assess models on Chess960. However:
   - In Ruoss et al., the model is constrained to produce legal moves; without this constraint, legal-move accuracy is extremely low (Table A3), making OOD analysis difficult.
   - In Schultz et al., the model is trained on Chess960 and then evaluated on Chess960; thus the evaluation is not OOD with respect to the training distribution.

   In contrast, we train only on standard chess and evaluate on several chess variants without any enforcement mechanisms, allowing us to directly probe intrinsic OOD generalization.
In summary, we believe that of the five contributions described in the paper, contributions (1), (3), (4), and (5) are novel, whereas contribution (2) is explicitly included as a reproduction of a chess transformer to allow the OOD study and is not the primary focus of our work.

We sincerely thank the reviewers for highlighting that these papers should be discussed in our manuscript. We firmly believe that our contribution — uncovering how transformer models exhibit compositional generalization and strategic adaptation — is not redundant with the two prior papers. Nevertheless, we take the feedback seriously and will revise the manuscript to clarify the distinctions and explicitly discuss the relationship to Ruoss et al. and Schultz et al. in the related works section.

---

### Note · Authors · 2025-12-01

**Comment:**

We thank the reviewers again for their thoughtful feedback. Although we believe our work offers novel empirical insights into transformer reasoning and compositional generalization, the concerns raised—particularly regarding its positioning relative to recent papers—indicate that substantial revision is needed in the narrative. We have therefore decided to withdraw the submission and will revise the manuscript before resubmitting to a future venue.

**Withdrawal Confirmation:**

I have read and agree with the venue's withdrawal policy on behalf of myself and my co-authors.